# Digital Interventions for Psychological Comorbidities in Chronic Diseases—A Systematic Review

**DOI:** 10.3390/jpm11010030

**Published:** 2021-01-06

**Authors:** Marta Maisto, Barbara Diana, Sonia Di Tella, Marta Matamala-Gomez, Jessica Isbely Montana, Federica Rossetto, Petar Aleksandrov Mavrodiev, Cesare Cavalera, Valeria Blasi, Fabrizia Mantovani, Francesca Baglio, Olivia Realdon

**Affiliations:** 1Center for Studies in Communication Sciences “Luigi Anolli”, Department of Human Sciences for Education “Riccardo Massa”, University of Milano-Bicocca, Piazza dell’Ateneo Nuovo 1, 20126 Milan, Italy; m.maisto2@campus.unimib.it (M.M.); barbara.diana@unimib.it (B.D.); marta.matamalagomez@unimib.it (M.M.-G.); jessica.montana@unimib.it (J.I.M.); petar.mavrodiev@unimib.it (P.A.M.); fabrizia.mantovani@unimib.it (F.M.); 2IRCCS Fondazione don Carlo Gnocchi ONLUS, Via Capecelatro 66, 20148 Milan, Italy; sditella@dongnocchi.it (S.D.T.); frossetto@dongnocchi.it (F.R.); vblasi@dongnocchi.it (V.B.); fbaglio@dongnocchi.it (F.B.); 3Department of Psychology, Catholic University of the Sacred Heart, Via Nirone 15, 20123 Milan, Italy; cesare.cavalera@gmail.com

**Keywords:** telemedicine, digital health, chronic diseases, mental health, psychological comorbidities, depression, anxiety

## Abstract

Chronic diseases represent one of the main causes of death worldwide. The integration of digital solutions in clinical interventions is broadly diffused today; however, evidence on their efficacy in addressing psychological comorbidities of chronic diseases is sparse. This systematic review analyzes and synthesizes the evidence about the efficacy of digital interventions on psychological comorbidities outcomes of specific chronic diseases. According to the Preferred Reporting Items for Systematic Reviews and Meta-Analyses (PRISMA) guidelines, a systematic search of PubMed, PsycInfo, Scopus and Web of Science databases was conducted. Only Randomized Controlled Trials (RCTs) were considered and either depression or anxiety had to be assessed to match the selection criteria. Of the 7636 identified records, 17 matched the inclusion criteria: 9 digital interventions on diabetes, 4 on cardiovascular diseases, 3 on Chronic Obstructive Pulmonary Disease (COPD) and one on stroke. Of the 17 studies reviewed, 14 found digital interventions to be effective. Quantitative synthesis highlighted a moderate and significant overall effect of interventions on depression, while the effect on anxiety was small and non-significant. Design elements making digital interventions effective for psychological comorbidities of chronic diseases were singled out: (a) implementing a communication loop with patients and (b) providing disease-specific digital contents. This focus on “how” to design technologies can facilitate the translation of evidence into practice.

## 1. Introduction

Non Communicable Diseases (NCDs), generally known as chronic diseases, are defined as medical conditions that cannot be transmitted, being the result of a multifactorial combination of genetic, physiological, environmental and behavioral characteristics [1]. NCDs are collectively responsible for a large proportion of premature deaths and almost 70% of all deaths worldwide. They threaten progress towards the 2030 Agenda for Sustainable Development, which includes a target of reducing premature deaths from NCDs by one-third by 2030 [2,3,4].

Evidence suggests that mental illness plays a key role among chronic diseases risk factors [5,6,7]. Comorbid depression, in general, predicts the onset, progression, management and level of disability associated with the chronic disease [8,9]. Evidence shows that depression negatively influences cardiovascular outcomes [10] and is more frequent in patients with cardiovascular disease (CVD) [11,12]. Depression and anxiety often co-occur [13]; recent meta-analysis reported that 32% of people with cardiovascular diseases presented high anxiety levels and 13% of them had a declared anxiety disorder [14]. Moreover, a higher prevalence of comorbid depression and anxiety has also been documented among people with Chronic Obstructive Pulmonary Disease (COPD) [15,16,17] and diabetes [18,19].

An additional risk factor is that individuals with mental health conditions, such as anxiety and depression, are less likely to seek professional help for chronic diseases. Therefore, symptoms related to their psychological condition may affect adherence to treatment and medications as well as their prognosis [20,21]. Aimed at preventing and reducing chronic disease-related morbidity and mortality, interventions target psychological support, risk factors and medication adherence. Their implementation, however, can be difficult due to low-resource settings [22].

Currently, the emerging need for a sustainable management of chronic diseases represents one of the main objectives of digital health (DH) interventions [23,24,25]. The integration of digital health solutions (e.g., delivered via smartphones, tablets, and other smart technologies) in clinical interventions allows to better meet patients’ needs, foster engagement in the care process [26] and enhance the quality of healthcare [27] (for example, through tailored self-management programs). In fact, approaches based on digital health (DH) have become very popular in health care and public health services [28,29]. Current evidence shows that the advantages of using mobile health devices (*mHealth*) translate not only in the improvement of both diagnosis and treatment but also in a better social connection with people [30]. The facilitation of social connections can mitigate the worry and concerns about personal health and the health of family members (fatigue, irritability, fear and despair) [31] while, at the same time, positively influencing the subjective experience of one’s health challenges [32]. Digital solutions are nowadays integrated in interventions targeting patients with chronic diseases [33,34,35,36,37]. For instance, diabetes management usually includes physical activity monitoring and blood sugar home monitoring via *mHealth* systems [38,39,40] while heart failure interventions often comprehend monitoring of weight, symptoms and physical activity [41,42].

The data available from current literature on the effectiveness of digital interventions for chronic diseases are conflicting and the evidence for the efficacy of digital interventions in addressing psychological co-morbidities of chronic diseases is sparse. While some studies reported a potential efficacy [3] others yielded contradictory results [28]. The current covid-19 pandemic represents an additional urge to fill this knowledge gap. In fact, people living with a chronic disease are among the most vulnerable populations, having to face the difficult choice between risking Covid-19 exposure during face-to-face clinical appointments and postponing needed mental health support. Since a digital healthcare revolution is advocated to meet this challenge [43], clarifying how technologies are designed in effective digital interventions for psychological comorbidities in chronic diseases can substantially contribute to this aim.

The aim of this systematic review is to analyze and synthesize the evidence about the efficacy of digital interventions on the outcomes of psychological comorbidities (depression and anxiety) related to a specific group of chronic diseases in adult populations. Based on a recent systematic analysis [44] that places these chronic diseases as a top ten global leading cause of death, we focused on cardiovascular disease, stroke, chronic obstructive pulmonary disease (COPD), and diabetes.

## 2. Methods

### 2.1. Search Strategy

We conducted this systematic review according to the “Preferred Reporting Items for Systematic Reviews and Meta-Analyses (PRISMA)” guidelines and flow diagram. Bibliographical data were collected on 28 February 2020 including studies from 2010 to 2020, using PubMed, PsycInfo, Scopus and Web of Science databases. For each database we used the following combination of research keywords: (1) “digital technology “ OR “telemedicine” OR “internet intervention” OR “ehealth” OR “mhealth” AND “psychological comorbidities”; (2) “digital technology “ OR “telemedicine” OR “internet intervention” OR “ehealth” OR “mhealth” AND “anxiety”; (3) “digital technology “ OR “telemedicine” OR “internet intervention” OR “ehealth” OR “mhealth” AND “depression.” See detailed search strategy in Figure 1. Only full-texts were included in this research (e.g., conference papers were excluded), studies’ citations were retrieved independently for each string of keywords, across all databases. The first list of the collected studies was finally exported to Mendeley to remove duplicated items.

### 2.2. Study Eligibility and Selection

Bibliographical research was limited to the English language. The list of 7636 articles was imported to Rayyan [45] a web application for a semi-automatic initial screening of abstracts and titles, indicating inclusion or exclusion criteria for study selection. The studies had to match the following criteria for selection: (a) conducted on human participants (b) designed as randomized controlled trials (RCT); (c) include a digital intervention, for psychological comorbidities, directed to a selected chronic disease (cardiovascular disease, stroke, COPD, Diabetes); (d) include a digital intervention that assessed at least one psychological comorbidity (i.e., such as depression and anxiety), as a primary or secondary outcome measure; (e) include a digital intervention directed to a group of patients.

Studies in which the intervention was in support of other chronic diseases were excluded. Single case studies, pilot and feasibility studies and quasi-experimental studies were excluded (see Figure 2).

### 2.3. Data Extraction

As per the PRISMA guidelines, key information for each study was summarized using a data extraction sheet purposely built for this review. Five reviewers (MM, MMG, JIM, FR and PAM) analyzed independently the full texts and disagreements were resolved through consensus. Extracted data included: authors and publication date, clinical condition, patients’ characteristics, purpose of study, case vs. control group (size), control group (type), interventions, duration of assessments and time points, follow-up, outcome measures and overview of results on depression, anxiety and Quality of life outcomes (see Appendix A: Summary of studies reviewed).

### 2.4. Statistical Analysis

For the quantitative synthesis, we computed standardized mean difference (SMD) between experimental condition and control groups of change from baseline to immediately post-treatment. SMD was calculated as Hedges g with a 95% confidence interval (CI) for each outcome measure. First, we considered psychological comorbidities outcome measures for each study (overall effect), in order to evaluate the efficacy of digital interventions on depression and anxiety. In order to provide further specific analysis, we conducted other separate meta-analysis for each chronic disease (cardiovascular disease, COPD, diabetes).

For the effect size computation, the number of subjects was included and this corresponded to the number of enrolled participants in case of intention to treat analysis and to the number of subjects that completed the study in case of no intention to treat approach. For the overall effect, the mean SMD of all outcomes in each study and the variance from each study were pooled using two random-effects models, one for the domain of depression and one for the domain of anxiety. We used random-effects models to calculate effect sizes given the heterogeneity of the studies. The chronic disease specific effects (separately for cardiovascular disease, COPD, diabetes) were analyzed using a similar method. For effect calculations, correction for intercorrelation among outcomes was assumed at 0.7, according to procedures suggested by Rosenthal [46]. In general, negative values suggest a more considerable improvement in the experimental group than in the control one. Effect size g can be interpreted (in absolute terms) using suggestions by Higgins et al. [47], with g ≤ 0.30 indicating a small effect, g > 0.30 a medium effect and g > 0.60 a large effect, respectively. I^2^ statistic was used with 95% CI to count the proportion of actual variance from total observed variance (I^2^ values of 25%, 50% and 75% indicate low, moderate and large proportions of variance from the exact effect size, Higgins et al. [47]). The publication bias was assessed through the funnel plot, exploring studies dispersed around either side of the mean effect size. The “Trim and Fill” procedure [48] was utilized to evaluate missing studies that are likely to fall to make the plot symmetrical.

Statistical analyses were computed using R software, adopting the metaphor R package.

## 3. Results

### 3.1. Studies Included

The search strategy identified 12.544 studies; after removing duplicates, a total of 7636 articles were included for title and abstract screening into the Rayyan software. 40 studies were selected for full text analyses and 17 matched the inclusion criteria (see Figure 2). They included nine digital interventions on diabetes, four on cardiovascular diseases, three on COPD and one on stroke. They were all conducted in developed countries. Fourteen out of the seventeen studies selected for this systematic review found digital interventions to be effective in reducing depression and anxiety outcomes [49,50,51,52,53,54,55,56,57,58,59,60,61,62].

### 3.2. Risk of Bias

Risk of bias (see Appendix A: Risk of Bias) was calculated following the guideline of the “Cochrane Collaboration risk of bias tool”, according to the latest version (RoB2) statement [63]. A low risk of bias was assessed in fourteen [49,50,51,52,53,55,56,58,59,61,62,64,65,66] of the seventeen articles presented in this work. Only three studies [54,57,60] presented some concerns due to deviations from the intended interventions [57], randomization and allocation process [54] or missing outcome data [60]. Overall, results from this analysis showed a methodological solidity for the articles selected adequate to allow drawing meaningful conclusions from further analyses.

### 3.3. Interventions’ Description

A detailed description of interventions is included (see Appendix A: Interventions’ description and design). We extracted information about the specifics of the digital interventions in the experimental groups and the interventions in the control groups (if an active control comparator was included); the design of all the interventions, with specifics about the digital contents, digital tools and communication pathways designed and implemented.

Seven of the studies considered in this systematic review [50,51,52,56,59,60,61] proposed integrated digital interventions, accessible through an internet-enabled mobile phone, tablet or computer. They generally included disease-specific psycho-educational contents (recommendations and suggestions about lifestyle, physical activity and diet) and prevention programs, as well as self-management modules for goal setting and monitoring. All of the aforementioned interventions included Cognitive Behavioral Therapy (CBT) modules, such as behavioral activation (BA) activity scheduling, problem solving, graded task assignment (GTA), relaxation, cognitive restructuring and relapse prevention.

Three studies [54,55,57] proposed remote psychotherapy treatments including at least one CBT module. These were administered to patients by trained healthcare professionals, via in-home video conferencing [55] or telephone/mobile [54,57]. Four studies [58,62,64,66] proposed telemonitoring interventions, with or without standard care. In the intervention by Lewis and colleagues [62] in addition to standard care, patients had to record physical symptoms (i.e., chest condition) at home through a handheld telemonitor connected via an ordinary telephone line. After seven consecutive days without data upload, the clinicians called the patients by phone. In the study by Pinnock and colleagues [66] patients were equipped with a tablet to record vital parameters and symptoms; the clinical team could monitor data online and contact patients by phone in case of issues with questionnaire completion or out-of-range parameters.

In the Baron and colleagues’ study [64] patients had to respond to daily questionnaires about their symptoms, habits and mood via their mobile phones. Stored and transmitted to a server, these data were accessible by both patients and nurses via a web portal. Patients received feedback both automatically (color-coded graphical display after data transfer) and from clinicians (recommendation to contact diabetes specialist nurses in case of out-of-range clinical readings).

The intervention by Wayne and colleagues [58] was based on a public web platform that allowed clinicians to monitor patients’ progress online and give support. Patients could message or do phone calls H24 with clinicians.

Two studies [53,65] focused their interventions on disease-specific rehabilitation through tailored digital contents for physical activity. The study of Antypas and colleagues [65] was web-based, providing reminders of scheduled activities and feedback on goals. The intervention implemented by Vloothuis and colleagues [53] consisted of eight weeks of tailored exercise therapy, for the patient and the caregiver, delivered through an e-health application.

The last study considered in this review [49] proposed an intervention designed for patients with Coronary Heart Disease (CHD), delivered via text messaging on mobile phones. Contents for the four modules (smoking, diet, physical activity and cardiovascular health) were tailored according to baseline assessment and a fully automated and personalized message system, which provided information, advice and support.

### 3.4. Treatment Efficacy on Psychological Comorbidities

A detailed summary of the interventions considered is provided in Appendix A: Interventions’ description and design. Fourteen out of the seventeen articles considered, found digital interventions to be effective in reducing depression and anxiety outcomes [49,50,51,52,53,54,55,56,57,58,59,60,61,62].

Ten interventions proved more effective than usual care treatments. Considering their digital intervention design (see Appendix A: Interventions’ description and design), the communication exchange implemented can be synthesized as an interactive two-way loop [50,54,56,57] a combination of automated and in vivo interactions [49,50,57] or a simpler clinician-to-patient pathway [59,60,61].

For seven out of these ten, the intervention included a remote psychological treatment [50,52,56,59,60] sometimes delivered by phone [54,57].

When incorporated in the intervention, the content design that proved more effective than usual care was always disease-specific and provided reminders of the intervention activities over the treatment period. Disease-specific modules were either combined with a remote psychological treatment [50,52,56,57,59] Two studies [53,61] were tailored according to in-itinere assessment and one study [60] to baseline assessment.

Altogether, results show that the clinician-to-patient pathway (via text messages or through direct contact with clinicians) is the minimally sufficient attribute for an efficient intervention design. Strong predictors of efficacy can be identified in (a) communication loop (even in phone-based psychotherapy treatments) and/or (b) digital contents designed to address specific disease characteristics.

The efficacy of disease-specific contents was apparent also in studies [50,54,59] with active comparators delivering digital non disease-specific interventions (i.e., befriending sessions). At least one of the predictors mentioned above was incorporated in studies that proved as effective as active controls [51,54,55]. Consistently, interventions focusing exclusively on telemonitoring [66] were found not effective.

### 3.5. Results of the Meta-Analysis

To test the efficacy of digital interventions on psychological comorbidities (depression and anxiety) we included 11 studies [50,51,52,53,54,56,58,59,61,64,66] in the meta-analysis. Six studies were excluded for missing/incomplete data.

The overall effect of digital interventions on depression outcomes was medium and statistically significant (g = −0.37; 95% CI [−0.60, −0.14]; *p* = 0.002). True heterogeneity across studies was large (I^2^ = 89.15%; Q = 92.64; df = 10; *p* < 0.001). The funnel plot showed a slight asymmetry. The Trim and Fill method suggests that no additional studies would be requested to make the plot symmetric (see Figure 3—Forest plots panel A for the global effect and Figure 3—Forest plots panels C, E and G for chronic disease specific effects. For more details, see Appendix A—Funnel plots).

The overall effect of digital interventions on anxiety, calculated from 8 studies, was small and non-significant (g = −0.17; 95% CI [−0.41, 0.08]; *p* = 0.178). True heterogeneity across studies was high (I^2^ = 90.25%; Q = 56.85; df = 7 *p* < 0.001). The funnel plot showed a mild asymmetry and that only one study provided a large effect size. The Trim and Fill procedure proposes that no further studies would be necessary to make the plot symmetric (see Figure 3; in particular: Forest plots panel B for the global effect and Forest plots panels D, F and H for chronic disease specific effects. For more details, see Appendix A: Funnel plots).

## 4. Discussion

The aim of this systematic review was to analyze and synthesize the evidence about the efficacy of digital interventions on the outcomes of specific chronic disease -related psychological comorbidities (depression and anxiety) in adult populations.

Seventeen RCTs were selected for this review; fourteen studies found digital interventions at least as effective as the control condition. The meta-analysis revealed an overall moderate and significant effect on the depression outcome. Considering the specific chronic diseases included, persons with diabetes and COPD benefited from interventions more than those suffering from cardiovascular disease or stroke. Although the effect on anxiety was small and non-significant, the same pattern of results was partially replicated regarding the specific chronic conditions, with persons suffering from diabetes lowering their anxiety symptoms more than people with cardiovascular diseases. Notably, analyses detected that the true heterogeneity across studies treating both depression and anxiety outcomes was large. To this respect, it is noteworthy that study protocols of the included researches were designed with control conditions consisting, in some cases, in usual care treatments, and, in other cases, in active control comparators. Moreover, the digital solutions implemented ranged from the adoption of technological devices for synchronous communication (i.e., phone-based treatments) to system architectures providing digital contents in an asynchronous fashion. Therefore, evidence on the design of digital solutions can play a pivotal role in the set-up of study protocols of digital interventions, both by pushing the need to employ active comparators as control conditions and to collect evidence on the efficacy of specific design solutions (i.e., synchronous vs. asynchronous options).

Greenwood and colleagues (2017) [67] highlighted that, in the management of chronic conditions, an iterative and complete feedback loop including monitoring and interpretation of data, subsequent adjustment of treatment and two-way communication between clinician and patient, is an essential component of an effective digital intervention [67]. Recently, Di Tella and colleagues [68] proposed the term Integrated Telerehabilitation Approach (ITA) to refer to rehabilitative care beyond the hospital setting in which technology allowed for the double communication loop between the hospital and the patient. In the same vein of Isernia and co-workers [69], they state that a double communication loop has a pivotal role in effective telerehabilitation since it permits both the planning and the adjustment over time of individualized patient-centered interventions. In fact, implementing a complete communication loop in the digital intervention was a crucial design option for achieving efficacy in some of the studies considered for this review and the clinician-to-patient pathway was a basic requirement for all interventions that proved to be effective. These results are in line with the Positive Technology (PT) framework [70] that views technologies as interfaces for personal experience, rather than as technological devices and points out the ultrasocial and hyper-cooperative characteristic of patients as humans [71]. Results from this review also suggest that treatment efficacy on psychological comorbidities can be reached even with the implementation of minimum technological requirements (i.e., phone-based psychological treatments). Also recent studies evidenced that a remote psychological treatment (CBT) aimed at reducing depression and anxiety symptoms in adults with depressive symptoms, delivered by phone, was as effective as the same treatment administered face-to-face [72,73,74,75,76]. Concerning the design of the digital content, this work shows that interventions based on the administration of digital contents are effective when they are designed to target the specific chronic disease treated. In addition, the tailoring of disease-specific content (at baseline and/or in itinere, based on patient’s profile and/or performance) can be considered as an enhancing factor of efficacy, in line with a recent work highlighting that the personalization of contents and the timing of the intervention delivery are usage-facilitators for mobile technology [77]. We found that digital interventions consisting solely in a telemonitoring component proved not to be effective. Indeed, as documented by Jimison and colleagues (2008) [78] systems that deliver reminders, alone or based on patient self-monitoring, are not consistently effective, except when combined with tailored digital contents or a bidirectional information flow.

Translation of evidence into practice takes on great importance today with the outbreak of the novel coronavirus-caused respiratory disease (COVID-19) [79]. COVID-19 caused a global health crisis affecting the whole health system, due to its intrinsic characteristic as communicable disease and its capacity to negatively affect people living with a pre-existing chronic disease [80,81].

In the absence of a medical cure, isolation, early diagnosis, symptomatic monitoring of contacts, lockdowns and quarantines, represent the most immediate strategies to face the pandemic emergency. In this context, telehealth offers a wide range of digital technologies with the potential to expand access to services and enhance public health strategies. For example, some authors recently proposed solutions such “virtual clinics” for telemedicine consultations, with imaging data uploaded from linked sites and analyzed remotely. This way, patient’s continuity of care is guaranteed and at the same time the possibility of contagion in hospital waiting rooms is reduced to a minimum. AI-based triage systems have been proposed to reduce physicians’ clinical load: for example, online medical “chat bots” may help patients to identify early symptoms and cope with them. Additionally, digital applications could help in collecting patient’s data (i.e., daily temperature, physiological symptoms) preventing hospital consultations in case of mild symptoms [82]. The isolation and reduced social contacts (necessary to slow the spread of the virus) undermine the regular social support systems, causing loneliness and representing a consistent risk for mental health. In fact, several COVID-19 related psychological symptoms have been globally observed in the population (i.e., depressive symptoms, anxiety and paranoia); regarding this issue, digital solutions could help in supporting patients, family members and health services providers [83].

Currently, in many countries (especially United Kingdom and United States of America) telehealth has been promoted (i.e., video consultation) to reach patients at their own home [84]. Unfortunately, most countries do not provide a national health system that fully integrates and finances telehealth services, even during an emergency. The contingent strategy for healthcare services therefore leads to reducing many clinical services or postponing routine medical appointments [84]. These strategies are not sustainable in the long time, regarding in particular people living with chronic diseases, having they to face the difficult choice between risking Covid-19 exposure (during face-to-face clinical appointments) or postponing the needed mental health support.

COVID-19 health challenge urged the implementation of telehealth to face the emergency; within this context, a longer-term goal could be achieved. Design and develop helpful solutions today, could enhance the public and institutional acceptance of digital technologies, expanding areas of healthcare and access to care services [82].

To reach this goal, the World Health Organization has recently published recommendations on digital interventions for health systems strengthening [85]. Despite recognizing that digital technologies provide concrete opportunities to tackle health system challenges, they also acknowledge that the enthusiasm for digital health has led to a variety of design implementations and a notable diversity of digital tools. The main urge advanced regards, first of all, considering digital interventions not as substitutes for established functioning health services but as interventions that can enhance and complement care for people for people with chronic diseases beyond the hospital/clinical settings, that is reaching their homes. Decision makers, stakeholders and practitioners in the health domain are recommended to consider the implementation of digital interventions, in a scenario that includes a number of factors such as, for instance, the specific health domain area, (and associated content), the software and communication channels to deliver them and the opportunity to leverage them in a cohesive and multi-faceted approach (rather than operating as singular units). In considering the implementation of a digital intervention as a unit of analysis, attention is devoted to the opportunity to rely on evidence-based protocols that can inform how technology is incorporated in the design of intervention (i.e., being modeled according mainly asynchronous or online synchronous communication exchanges). The opportunity to design RCTs involving an active control comparator goes in this direction, since it allows to identify the “active ingredients” that can explain how specific technological options impact the efficacy of an intervention. Along this line, our focus was on the “how” of designing technologies targeting a specific disease condition, since the diversity of implementation efforts to design the communication channels (and, if the case, the related foundational layer of ICT) needs to match the resources available and the organizational impact within the local context in which it is implemented.

## 5. Conclusions

This systematic review has inherent strengths and limitations. First, only RCT studies were considered, guaranteeing methodological solidity; in fact, a low risk of bias was assessed in fourteen out of seventeen articles selected (see Appendix A: Risk of Bias). Nonetheless, a crucial limitation was inherent in the design of the studies; only six of them included an active control comparator. The choice of an active control comparator is critical for disentangling how the design of technology in psychological interventions contribute to their effectiveness.

Chronic diseases can be considered optimal target conditions for the development and implementation of telemedicine approaches [33]. The present work examined RCT studies with the aim of identifying the elements that made digital interventions effective for ameliorating psychological comorbidities of chronic diseases. We identified two main patterns of effective digital intervention design, ranging from the adoption of low-end and/or low-cost technologies (i.e., phones) to more resource-absorbing options (i.e., development of disease-specific digital contents on web platforms). These results can be quite useful in the translation of evidence into practice, since developing and implementing digital contents, and/or providing a complete communication loop through an application or system architecture can be quite effortful and the balance between pitfalls and opportunities needs to be accurately managed. This focus on “how” to design technologies matches with the roadmap envisioned for the future of telehealth [86] especially during and after the covid-19 pandemic [43].

## Figures and Tables

**Figure 1 jpm-11-00030-f001:**
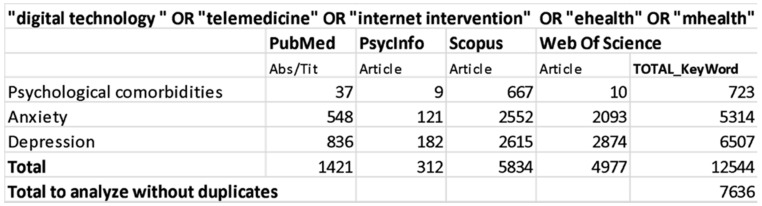
Data search strategy.

**Figure 2 jpm-11-00030-f002:**
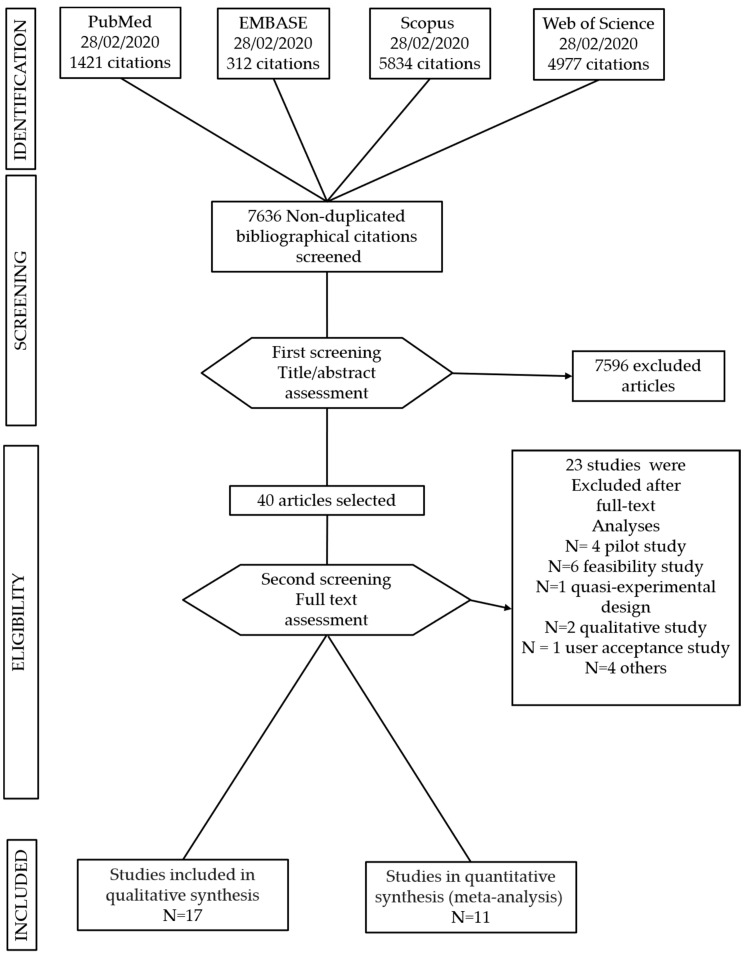
Preferred Reporting Items for Systematic Reviews and Meta-Analyses (PRISMA) flow chart of study selection.

**Figure 3 jpm-11-00030-f003:**
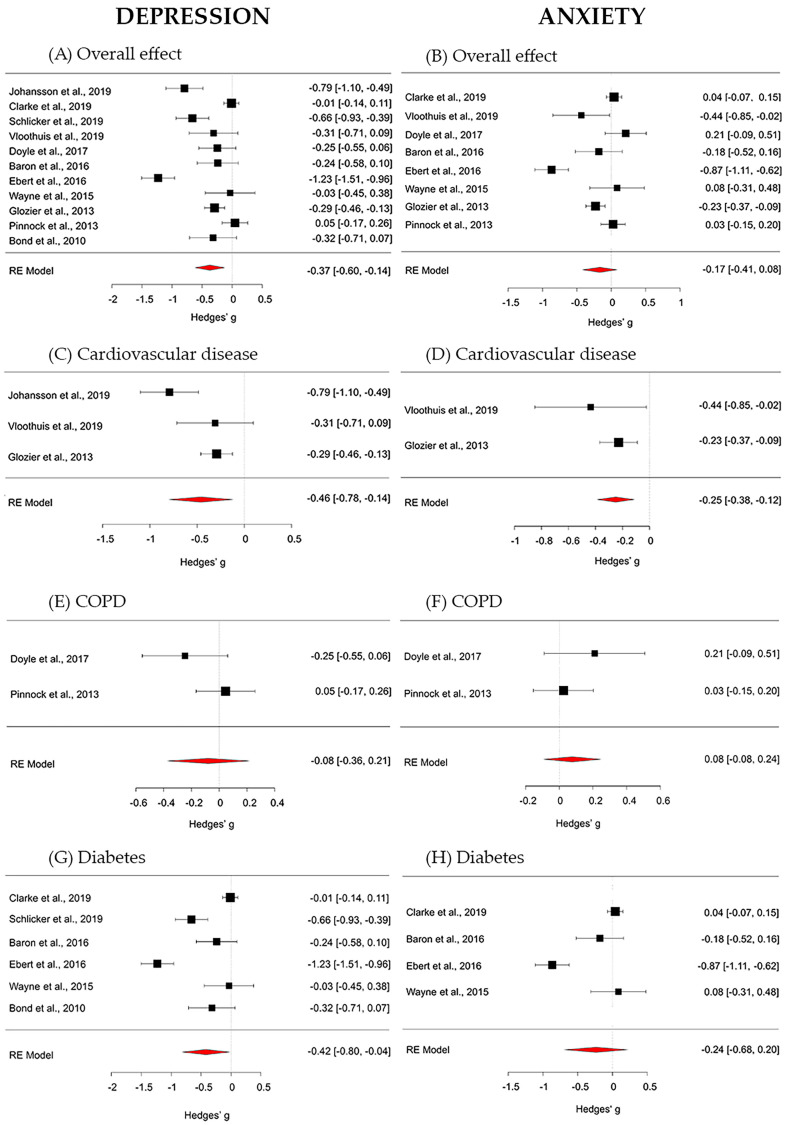
Forest plots on depression outcomes (panel **A**—overall effect; panel **C**—cardiovascular disease; panel **E**—chronic obstructive pulmonary (COPD) disease; panel **G**—diabetes) and on anxiety outcomes (panel **B**—overall effect; panel **D**—cardiovascular disease; panel **F**—COPD; panel **H**—diabetes).

## Data Availability

No new data were analyzed in this study.

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
