# Peer review of "Digital Interventions for Psychological Comorbidities in Chronic Diseases—A Systematic Review"

_jpm, 2021, doi:10.3390/jpm11010030_

Round 1

Reviewer 1 Report

Thank you for this review. I believe that the article can be published, I have no objections to the methodology and discussion of the results.  

Author Response

Thank you very much for your comments.

Reviewer 2 Report

Due November 4, 2020

Review request: Journal of Personalized Medicine

Type of manuscript: Review

Manuscript ID: jpm-987032

Title:    Digital interventions for psychological comorbidities in non-communicable diseases: a systematic review

Synopsis

A PRISMA guided systematic review was conducted on articles published from XXXX to XXXX using databases PubMed (Medline), PsycINFO (EBSCO), Scopus (Elsevier), and Web of Science (Clarivate Analytics) on the topic of “digital intervention” for depression and/or anxiety in patients with non-communicable diseases (NCD). The Rayyan platform was used to compile searched articles. The search identified 7,636 unique titles (articles, dissertations, theses, conference proceedings, etc.), and 40 titles were selected by consensus of three coauthors who reviewed and evaluated the abstract, title and/or summary. The full texts of the 40 articles were further analyzed and 23 studies were excluded for reasons specified in figure 2. The 17 remaining articles were summarized in the Table 1 spreadsheet. The type of diseases and number of articles were: 14 diabetes, 4 cardiovascular diseases, 3 COPD, and 1 stroke. Two elements that make digital interventions effective for psychological comorbidities of NCD were: 1) implementing a communication loop with patients, and 2) designing disease-specific digital content.     

Reviewer's conflict of interest: None

Comment to authors

Cancer is a NCD. Lancet states, “An estimated 32·2 million NCD deaths (80%) were due to cancers, cardiovascular diseases, chronic respiratory diseases, and diabetes, and another 8·3 million (20%) were from other NCDs.” (PMID: 30264707) When I performed a simple query on PubMed, I found PMID:28297187, a 2017 RCT study on internet-delivered CBT for cancer patients with depression and/or anxiety. This article was not included in this systematic review of NCD-related digital psychological interventions, which leads to some concern about missing publications. Specific comments are below:

  1. Title: The use of term “NCD” is discussed in later comments. Because the NCD burden of society differs from country to country, a geo-economical description of the majority of the 17 articles can be included, such as “EU region” or “developed countries.” I personally am considering the social burden of NCD in developing countries to be nutritional deficiency related.
  2. Abstract: Well summarized except the use of the term NCD
  3. Introduction: Well-presented except the use of the term NCD
  4. Methods:
    1. The date of the search (February 28, 2020) was specified; however, the range of publications searched was not indicated. The range of publications is a critical inclusion criteria.
    2. The Boolean terms (Line 89-93) do not include RCT. Because the Figure 2 PRISMA flow chart excluded 23 studies that are not RCTs, it was not clear where RCT has been selected. Please note that inclusion and NON-exclusion are different and this procedural change may result in different final selections.
    3. The alphabetical list in Lines 104-109 has typos.
    4. Line 106. Can you justify why cancer is not an NCD? If cancer was excluded because there were too many cancer-related articles to perform a practical systematic review study, the term “NCD” should not be used. Instead “four selected chronic diseases” may be appropriate. If the authors wish to keep this study publication as a systematic review of NCT-related psychological comorbidity (depression and anxiety) and its digital intervention, cancer-related articles must be included. Operationally defining the NCD but excluding cancer may confuse other researchers using NCD as a Boolean search term.
  5. Results:
    1. Not sure about the contribution of bias risk analysis (Table 2). The main concern was whether or not the methods provide an exhaustive list of appropriate articles.
    2. Table 3, “Intervention’s description and design” is the most important data source in this study for qualitatively synthesizing two findings: “1) implementing a communication loop with patient, and 2) designing disease-specific digital contents.” Report of the analysis of this table was limited to the description and treatment efficacy sections. Unfortunately, these two sections do not convey the finesse of qualitative analysis. Briefly, qualitative methods require: 1) a methodological perspectives, 2) an interpretive lens, 3) some sort of analysis technique such as coding (Line 151 mentioned “communication loop” was coded), 4) a method of consensus analysis by different coauthors, and 5) description of results which may be quantitative, such as in line 194 “fourteen out of the seventeen” or line 200 “seven out of these ten.” However, this study did not seem to conduct a formal qualitative synthesis; thus, this comment applies to future studies by these investigators.
    3. This reviewer was unable to read the 17 selected articles; however, from first impressions, it was surprising that they did not contain statistical values of depression or anxiety inventories in each RCT study. The purpose of RCT studies is to evaluate quantitative values (effect size), and this study does not report a statistical value for the evaluation of efficacy.
  6. Discussion: No comment
  7. Conclusion: No comment

End of review.

Author Response

  • Title: The use of term “NCD” is discussed in later comments. Because the NCD burden of society differs from country to country, a geo-economical description of the majority of the 17 articles can be included, such as “EU region” or “developed countries.” I personally am considering the social burden of NCD in developing countries to be nutritional deficiency related.

We completely agree with the fact that the NCD burden differs from country to country. In fact, all our selected articles refer to interventions implemented in developed countries (Norway, Australia, Sweden, UK, USA, Germany, Netherlands, Canada). We therefore substituted the term “NCD” in the title with “chronic diseases”. However, we did not add the geo-economical classification (“developed countries”) since, after a PubMed search, we saw that it was more frequent in relation to NCDs - but not to chronic diseases. We added the geo-economical classification of the included interventions in the manuscript (“developed countries”) in lines 135-136.

  • Abstract: Well summarized except the use of the term NCD

Thank you for your comment. Regard the use of the term NCDs, as you can see in line 17, we replaced it with the term “chronic diseases”.

  • Well-presented except the use of the term NCD

Thank you again for your comment. When necessary, we replaced the term NCD with “chronic disease” throughout the manuscript. 

  • Methods:
  • The date of the search (February 28, 2020) was specified; however, the range of publications searched was not indicated. The range of publications is a critical inclusion criteria.

Thank you for this observation. We followed your suggestion and added the date of search (please see line 92).

  • The Boolean terms (Line 89-93) do not include RCT. Because the Figure 2 PRISMA flow chart excluded 23 studies that are not RCTs, it was not clear where RCT has been selected. Please note that inclusion and NON-exclusion are different and this procedural change may result in different final selections.

Thank you for this observation; it allows us to better explain our selection criteria process. We selected the RCT studies during the title and abstract screening phase, using Rayyan, a free web application built to faster this phase with a semi-automatic process. In the revised manuscript, we better explained how it works to clarify our selection criteria (please see lines 108-109).

  • The alphabetical list in Lines 104-109 has typos.

Thank you for the indication, we fixed the alphabetical list (lines 110-115).

  • Line 106. Can you justify why cancer is not an NCD? If cancer was excluded because there were too many cancer-related articles to perform a practical systematic review study, the term “NCD” should not be used. Instead “four selected chronic diseases” may be appropriate. If the authors wish to keep this study publication as a systematic review of NCT-related psychological comorbidity (depression and anxiety) and its digital intervention, cancer-related articles must be included. Operationally defining the NCD but excluding cancer may confuse other researchers using NCD as a Boolean search term.

Thanks to your observations, we had the opportunity to improve the manuscript, hoping it could be clearer and more accessible to readers. Regarding the use of the NCD term, we agree with you that it may be confounding using a general term to indicate only a group of NCDs excluding cancer. We decided to not include cancer and focus only on cardiovascular disease, stroke, chronic obstructive pulmonary (COPD) disease, and diabetes because a recent systematic analysis (see line 84-85) placed these NCDs as a top ten global leading cause of death. Please note that we did not use NCD as a Boolean search term; we used it only as an “indicator” in the inclusion criteria set on Ryyan.

Following your comments, we replaced “NCD” with the more appropriate “chronic disease” in all the sections of the revised manuscript (title, abstract, introduction, methods, discussion); furthermore, we specified in brackets which type of chronic disease were selected (line 112).

  • Results:
  • Not sure about the contribution of bias risk analysis (Table 2). The main concern was whether or not the methods provide an exhaustive list of appropriate articles.

We assessed the risk of bias of the 17 studies selected for this review, according to Cochrane guidelines for assessing risk of bias in RCTs, considering seven evidence-based domains. The main concern was the methodological solidity of the selected articles, in order, for this review, to be able to draw useful conclusions. Following your suggestion, we better clarified in the revised manuscript the aim and contribution of risk of bias assessment for this work (see lines 149-150).

  • Table 3, “Intervention’s description and design” is the most important data source in this study for qualitatively synthesizing two findings: “1) implementing a communication loop with patient, and 2) designing disease-specific digital contents.” Report of the analysis of this table was limited to the description and treatment efficacy sections. Unfortunately, these two sections do not convey the finesse of qualitative analysis. Briefly, qualitative methods require: 1) a methodological perspectives, 2) an interpretive lens, 3) some sort of analysis technique such as coding (Line 151 mentioned “communication loop” was coded), 4) a method of consensus analysis by different coauthors, and 5) description of results which may be quantitative, such as in line 194 “fourteen out of the seventeen” or line 200 “seven out of these ten.” However, this study did not seem to conduct a formal qualitative synthesis; thus, this comment applies to future studies by these investigators.

Thanks for your suggestions; we will certainly take into account a more formal qualitative analysis for our next review work. In Table 3 (Interventions' description and design), we reported the communication pathways included in the interventions as extracted from the articles; we then synthesized recurring elements and labeled “two-way loop” the situation in which the exchange of information was allowed by the system in both directions (patient-to-clinician and clinician-to-patient). Some changes were done in the revised manuscript to better clarify this process (see lines 205-206).

  • This reviewer was unable to read the 17 selected articles; however, from first impressions, it was surprising that they did not contain statistical values of depression or anxiety inventories in each RCT study. The purpose of RCT studies is to evaluate quantitative values (effect size), and this study does not report a statistical value for the evaluation of efficacy.

Thanks for this comment. We reported the statistics of results in the revised Table 1 (Summary of studies reviewed, last column), as retrieved in the articles. Of course some kind of metanalysis would have improved the quality of this work but it was beyond the scope of our research and impossible to complete and include, within deadline, in the present manuscript.

Reviewer 3 Report

This review is timely prepared, and very well constructed , since this kind of work is needed to cope with covid-19 , I suggest to the authors to expand the discussion part and elaborate more how telehealth could help the health providers to outreach the public and provide the essential services during lockdown from covid-19.

I recommend also to highlight the global health policies on using Digital interventions  for psychological comorbidities in NonCommunicable Diseases such as WHO and CDC....; putting clear recommendations at the last section as well as on the abstract.

Author Response

  • This review is timely prepared, and very well constructed , since this kind of work is needed to cope with covid-19 , I suggest to the authors to expand the discussion part and elaborate more how telehealth could help the health providers to outreach the public and provide the essential services during lockdown from covid-19.

Thank you for this valuable suggestion. According to your comment, we expanded the discussion; please see the changes the revised manuscript (lines 264 to 295)

  • I recommend also to highlight the global health policies on using Digital interventions for psychological comorbidities in NonCommunicable Diseases such as WHO and CDC....; putting clear recommendations at the last section as well as on the abstract.

Thank you again. We added in the Abstract the Who recommendation on considering Digital Health interventions primarily as enhancers (and not as substitutes) of established non-digital Health Systems. We also elaborated on WHO recommendations about the contribution of digital interventions for health system strengthening in the Discussion (lines 296-317).

Round 2

Reviewer 2 Report

Dear Editor and Authors:

The research design, including the qualitative data extraction, requires further upgrading. As the author noted, this manuscript needs to be combined with meta-analysis because the articles selected were chosen to perform meta-analysis with; however, the authors instead extracted qualitative information from these articles. There are also discrepancies in the fundamental research design.

A methodological flaw is difficult to modify IF the research is the original primary study. This is a review, not a primary study thus their flaw can be edited by: 1) change inclusion and exclusion criteria for the purpose of qualitative information extraction, or 2) perform additional analysis, in this case meta-analysis. Their selected articles (RCT) are designed to do meta-analysis. I mentioned this on my first review. Author's reply was 'not enough time to perform an additional analysis'. Their attempt to meet the criticism was to add the raw statistics for each study in the summary table. They may need additional statistician's help; however, meta-analysis calculation is necessary for this manuscript to stand scientific scrutiny.

Give more time and let them perform the additional analysis.

Author Response

We sincerely thank you for encouraging us to upgrade our research adding a quantitative synthesis.

Although we could include in the meta-analysis only 11 (out of 17) studies for missing/incomplete data (Authors of these papers did not answer our requests), we could perform analyses on eleven studies (for the depression outcome) and on eight studies (for the anxiety outcome).

We consistently updated the manuscript as follows:

  • we modified the Abstract stating that a quantitative synthesis was provided for psychological comorbidities outcomes (depression and anxiety)
  • in the Methods section, we added a paragraph in which we described the statistical analyses conducted (2.4 Statistical analysis)
  • in the Results section,we presented the results of the meta-analysis both as overall effect and regarding the specific chronic conditions in the studies included (3.5 Results of the meta-analysis). Moreover, we added i) a figure with the forest plots both for the global effect and for the chronic conditions specific effects (Figure 3) and ii) the funnel plots (Figure S1)
  • in the Discussion section, just after stating that 14 out of the seventeen studies included were at least as effective as the control condition, we discussed the results of the quantitative synthesis
  • we modified Table S1: Summary of the studies reviewed by changing the last column (formerly named Results on psychological comorbidities). We labelled this column: "Overview of results on depression, anxiety and QoL outcomes"  and included only statements of efficacy (no metrics) as presented by Authors in their respective papers. 

We are really grateful for your involvement and useful suggestion - and hope that the manuscript will have improved in quality, readability, and robustness of methodology.